# Assessment of the Content of Glycoalkaloids in Potato Snacks Made from Colored Potatoes, Resulting from the Action of Organic Acids and Thermal Processing

**DOI:** 10.3390/foods13111712

**Published:** 2024-05-29

**Authors:** Anna Pęksa, Agnieszka Tajner-Czopek, Artur Gryszkin, Joanna Miedzianka, Elżbieta Rytel, Szymon Wolny

**Affiliations:** Department of Food Storage and Technology, Faculty of Biotechnology and Food Science, Wrocław University of Environmental and Life Sciences, Chełmońskiego St. 37, 51-630 Wrocław, Poland; anna.peksa@upwr.edu.pl (A.P.); artur.gryszkin@upwr.edu.pl (A.G.); joanna.miedzianka@upwr.edu.pl (J.M.); elzbieta.rytel@upwr.edu.pl (E.R.); szymon.wolny@upwr.edu.pl (S.W.)

**Keywords:** red- and purple-fleshed potatoes, snacks, French fries, glycoalkaloids, organic acids, microwaving, frying, baking

## Abstract

Glycoalkaloids (TGAs, total glycoalkaloids), toxic secondary metabolites, are found in potatoes (110–335 mg·kg^−1^ DW), mainly in the peel. Colorful, unpeeled potatoes are an innovative raw material for the production of snacks which are poorly tested in terms of their glycoalkaloid content. Third-generation snacks and French fries made from red-fleshed Mulberry Beauty (MB) and purple-fleshed Double Fun (DF) potatoes were produced with the use of 1% solutions of ascorbic, citric, lactic, malic, and tartaric acids to stabilize the structure of anthocyanins in the raw material and maintain their color in obtained products. The influence of the type of acid and thermal processes, like frying, microwaving, and baking, on the content of glycoalkaloids in ready-made products was examined. Only 0.45–1.26 mg·100 g^−1^ of TGA was found in pellet snacks and 1.32–1.71 mg·100 g^−1^ in French fries. Soaking blanched potatoes in organic acid solution reduced the α-chaconine content by 91–97% in snacks and by 57–93% in French fries in relation to the raw material to the greatest extent after the use of malic acid and the DF variety. The effect of lactic and citric acid was also beneficial, especially in the production of baked French fries from MB potatoes.

## 1. Introduction

Colored potatoes, as shown by various authors all over the world [1,2,3,4], are characterized by increased attractiveness due to the unusual color that distinguishes them from the traditional yellow, cream, or white varieties that is given to the products obtained from them. They are also a potential source of health-promoting ingredients, including anthocyanins, representing numerous polyphenolic compounds found in potatoes [1,2,5,6,7,8]. Many studies are being carried out on increasing the amount or stabilizing the structure and activity of these ingredients in potatoes and their products, both for health reasons and sensory attractiveness. The importance of anthocyanins as active substances with antioxidant and free radical scavenging properties is emphasized, including their importance in inhibiting fat oxidation processes in fried potato products [2,9].

Potatoes of varieties with colored flesh, like those with traditional flesh color, contain glycoalkaloids produced in tubers as secondary metabolites. These compounds are represented in potatoes mainly by α-solanine and α-chaconine and exhibit toxic effects when their total amount reaches approximately 200 mg·kg^−1^ fresh weight FW, causing neurological disorders such as apathy, drowsiness, and disorientation, and may even be fatal [10,11,12,13]. However, few studies also indicate the beneficial effects of small amounts of glycoalkaloids, including antipyretic and anti-inflammatory effects [2,10,14]. Research by these authors also shows that α-chaconine is a more toxic compound. Its content in potato tubers is 2–5 times higher than that of α-solanine. The total content of glycoalkaloids in potatoes ranges from 85 to 182 mg·kg^−1^ DM [2,10,11,14], and they are usually located at a distance of 1–1.5 mm from the outer part of the tuber [15]. The literature on the subject indicates that potatoes intended for food should not contain more than 12 mg·100 g^−1^ FW of total glycoalkaloids (TGA), as they are considered bitter and unacceptable to consumers. The result of research conducted by various research centers are potato varieties in which the accumulation of these compounds does not exceed 10 mg of glycoalkaloids in 100 g of tuber. Varieties with low TGA content also include, to a large extent, varieties with colored flesh [10,16,17,18].

Potato products, such as selected assortments of French fries and crisps, obtained from unpeeled tubers, contain more polyphenolic compounds, proteins and dietary fiber compared to products from peeled tubers, and the production process is more efficient and profitable. In addition, leaving the skin on the surface of tubers with colored skin significantly increases the content of anthocyanins in the products obtained from them [15]. However, it should be taken into account that attractive for consumers, products with a skin are also characterized by an increased dry matter, and thus a potentially higher share of glycoalkaloids in the product [2,7,15,17].

The peeling and blanching steps during potato processing allows for the removal of approximately 90% of the glycoalkaloids found in the tubers [15,19]. The blanching procedure, commonly used in potato processing, is supposed to have a positive effect on the color of the products and its uniformity, but also on the consistency, especially of French fries. According to science and practice, inorganic acids, such as sulfurous and organic acids, such as citric and ascorbic acids, used in the production of potato products, prevent the browning of potato tissues, i.e., the effect of the Maillard reaction [20,21]. When processing potatoes with red and purple flesh, the color of the finished products is also related to the transformation of anthocyanins that occur under the influence of high temperature [13,20,21]. In that conditions anthocyanins transform into colourless chalcones, which, when oxidized, can form high-molecular coloured compounds [22]. Due to the large number of factors, the mechanism of these transformations is not fully known. Anthocyanins have a high ability to attach numerous hydroxyl and methoxy groups in the flavylium cation ring. The degree of glycosylation and acylation of anthocyanidin with phenolic acids or organic acids improves the stability of anthocyanin pigments [23].

The literature on the subject lacks information on the influence of various organic acids on the degree and direction of stabilization of the structure and color of various acylated anthocyanins found in colored potatoes, stable in a wide pH range. Potato varieties with colored flesh are attracting the interest of an increasingly wide group of consumers in various countries around the world. Therefore, they are increasingly the subject of detailed research in terms of the possibility of their industrial processing. Due to the need to develop research on the properties of such potatoes an attempt was made to determine the effect of treating tubers with skin with selected food organic acids on the content of glycoalkaloids in dried potato grits, third-generation snacks made from it and in French fries in the process of frying, microwave oven or baking. The research used ascorbic, citric, lactic, malic, and tartaric acids, which in pilot studies had a positive effect on the color of potato products with purple and red flesh.

Therefore, the aim of the research was to determine the influence of the type of organic acid, the method of preparing products for consumption and the type of raw material, i.e., unpeeled potatoes of 2 varieties, differing in the color of the flesh, and thus in the structure and content of anthocyanins, on the content of glycoalkaloids in dried potatoes and third-generation snacks, and fries.

## 2. Materials and Methods

### 2.1. Raw Materials Characteristic

Samples of tubers of two potato varieties with colored flesh, red Mulberry Beauty (MB) and purple Double Fun (DF), from the 2021/2022 growing season, purchased directly from the producer in Poland constituted the research material. The tested potatoes were harvested in full maturity, without mechanical damage, greening, and infestation and without sprouts. The size of the potato tubers of MB variety was closed to 60 × 80 mm (width × length), and the tubers of the DF variety to 60 × 90 mm (Appendix A). In total, 20 kg of raw material of each variety was taken for the experiment potatoes collected for the study and then divided. Potatoes collected for the studies were divided into raw materials intended for obtaining freeze-dried potatoes (experimental grits), used in the course of research as a component of pellet snacks, and intended for the production of French fries. Five organic acids of analytical grade were used in the research: citric acid, lactic acid, L-ascorbic acid, malic acid, and L-tartaric acid. In the production of pellet snacks, apart from the experimental dried potatoes, potato starch obtained from a starch factory in Niechlów, as well as corn grits produced by Sante, Poland; and salt (NaCl) by Kłodawa, Poland, were also used.

### 2.2. Basic Analyses of Raw Materials and Ready-to-Eat Products

The basic chemical composition and a sum of anthocyanins were determined in the raw material. Determinations of the dry matter of fresh, unpeeled potato, freeze-dried raw material, ready French fries, and pellet snacks were carried out by the reduced weight until constant weight was achieved [24]. The starch content was determined in raw potato tubers indirectly by measuring the specific gravity of tubers and reading the starch value from Maercer’s tables, according to the methodology described by Houghland [25]. The reducing sugar was determined by the colorimetric method described by Lindsay [26]. The anthocyanin content was determined using the HPLC method, as described below. Freeze-dried potato samples were extracted with 70% aqueous acetone (acidified with 0.1% acetic acid). The homogenized mixture was left for 2 h at room temperature, and then the acetone–water mixture was separated with chloroform to remove lipophilic compounds. The acetone–water fraction was collected, and the remaining acetone was evaporated on a Büchi rotary evaporator. The obtained aqueous extract was brought to a known volume with distilled water and stored at 20 °C until analysis. Samples filtered through 0.2 µm filters were used for HPLC analysis of anthocyanins. Anthocyanins were determined using a Dionex HPLC system (Waltham, MA, USA) equipped with an Ultimate 3000 model diode detector, an LPG-3400A quaternary pump, an EWPS-3000SI automatic sampler, and a TCC-3000SD thermostabilized column chamber, controlled by Chromeleon v.6.8 software. An Atlantic T3 reversed-phase column (250 mm × 4.6 i.d., 5 μm) (Waters, Wexford, Ireland) and an Atlantis T3 guard column (20 × 4.6 i.d., 5 μm) (Waters Corp., Milford, MA, USA) were used. The following solvents were used as the mobile phase: A/4.5% formic acid and B/acetonitrile. The following elution conditions were used: 0–1 min, 5% B isocratic; 1–6 min, linear gradient from 5 and 10% B; 6–26 min, linear gradient 10–20% B; 26–33 min, linear gradient from 20 to 100% B; and then the initial conditions. The flow rate was 1 mL·min^−1^, and the injection volume was 40 μL. The column was operated at 30 °C. Anthocyanins were monitored at a wavelength of 520 nm.

### 2.3. Conditions for Producing Experimental Coloured Potato Grits

Washed, unpeeled potatoes intended for the preparation of experimental dried potatoes (in the form of grits of 0.5–1.0 mm) were cut into 1.0 cm slices using a mechanical slicer (Robot coupe CL 50, Vincennes, France). Then, they were blanched in water under conditions typical for French fries’ production, i.e., using water at a temperature of 75 °C and in 10 min. The hot slices were cooled in the ice water (2 min.), and then samples were separated and were immersed in parallel in water (control sample) and in 1% solutions of five organic acids, namely citric acid, lactic acid, L-ascorbic acid, malic acid, and L-tartaric acid, for 5 min. The potato slices obtained in this way were dried using the freeze-drying method, and the dried samples were grounded in a laboratory grinder (Retsch GM 200, Hann, Germany) and then passed through a sieve with a mesh size of 1 × 1 mm. The obtained dried potatoes in the form of grits were used as pellet ingredients. These were stored in the freezer before taking into the experiments. The scheme for preparing experimental potato grits, intermediate products, ready-made snacks, and French fries is shown in Figure 1.

### 2.4. Conditions for Obtaining Extruded Pellets and Colorful Snacks

To the mixture of potato starch (650 g·kg^−1^ of the mixture), experimental dried potato (260 g·kg^−1^ of the mixture), corn grits (50 g·kg^−1^ of the mixture), and salt (10 g·kg^−1^ of the mixture) water was added in an amount that the moisture of the obtained dough was around 35%. The proportions of the mixture for the extruded pellets were elaborated previously by the authors Pęksa et al. [27]. Then, the dough was passed through a sieve with a mesh size of 1 × 1 mm, packed in polyethylene bags, and kept at a temperature of 20 ± 2 °C for 24 h. After this time, the samples were rubbed to obtain uniform granulation and moisture, and then they were subjected to the extrusion process in a Brabender laboratory extruder, type 20 DN, using a screw with a compression ratio of 1:1, screw rotation speed of 120 rpm, a head with a die size of 80 × 0.5 mm, and the process temperature in three successive sections as 50–60–80 °C. The extruded product in the form of a strip was cut into pieces of 30 × 15 mm and dried at 20–22 °C to a moisture content of about 11%, i.e., for about 14–16 h. Ready semi-products (pellets) were stored for 1–2 days in tightly closed polyethylene bags at the room temperature until expanded snacks were obtained from them.

The pellet samples were divided into two batches, one for the preparation of snacks that expand during frying in oil, and the other for expansion under the influence of microwaves at 750 W. The microwave exposure time was set at 25 s through pilot studies based on sensory evaluation (results not included). The process of frying the pellets was carried out in rapeseed oil heated to the temperature of 180 °C, measuring the period of 3 s after the snacks floated to the surface of the oil. Samples of fried snacks were drained of excess oil on filter paper.

### 2.5. Conditions for Obtaining Coloured Potato French Fries

Unpeeled potato samples of the tested varieties, intended for the preparation of French fries, were washed, cut using a cutting machine (Robot Coupe CL50, Haas, Germany) into strips with a cross-section of 1 × 1 cm, and blanched in water at 75 °C for 10 min. Blanched cut potatoes were cooled in ice water for 2 min, and then the whole sample was divided into six parts and immersed for 5 min in parallel in water (control sample) and 1% solutions of five organic acids, i.e., citric acid, lactic acid, L-ascorbic acid, malic acid, and L-tartaric acid. Then, strips of experimental potatoes were drained on paper and subjected to two methods of thermal treatment (frying or baking). French fries were prepared using a two-step method. In the first stage, experimental samples of potato strips were fried in rapeseed oil at 175 °C for 1 min [28], and then they were cooled, frozen, and packed in polyethylene bags. These pre-fried frozen French fried potato samples were divided into two parts for final processing. One part was fried in rapeseed oil at a temperature of 175 °C for about 5 min (the proportion of oil to raw material was: 200 g·L^−1^ of oil), while the other part was thermally treated in a convection oven (Type SCC61 WE, Rational, Landsberg, Germany). For baking, samples of pre-fried French fries were placed on special baking trays and baked at 190 °C for 25 min. Samples of fried and baked ready-made French fries were degreased using the Soxhlet method and intended for determining the content of glycoalkaloids. The time and temperature of the thermal processing of potato strips in a convection oven were determined experimentally in pilot studies, based on the measurement of the dry weight of ready French fries and sensory evaluation (results not included).

### 2.6. Sample Preparation for the Chromatographic Analysis of α-Solanine and α-Chaconine

The freeze-dried samples of raw material and ready products, after grinding in a laboratory electric mill, constituted a fixed material in which the content of α-solanine and α-chaconine and the sum of these compounds (TGA) were determined. Samples of fried snacks and French fries were previously degreased in a Soxhlet apparatus. These were used for determining the content of glycoalkaloids based on the methodology provided by Saito et al. [29] and Pęksa et al. [30], with some modifications. A sample 5 g of freeze-dried material was mixed with 25 mL of methanol and placed in an ultrasound bath for 30 min, followed by filtration. The filtrate was brought to a final volume of 50 mL with methanol. An aliquot of 5 mL was mixed with 8 mL of water and cleaned up on the SPE column—Chromabond C18 ec.; 500 mg; 6.0 mL (Macherey-Nagel, Dueren, Germany). The eluate was evaporated to dryness under vacuum at a temperature of 50 °C, and dry residue was dissolved in 1 mL of methanol. The whole solution was filtered through syringe filter (Pureland, hydrophobic PTFE, 0.22 μm). Standard solutions (1 mg⋅mL^−1^) were prepared by dissolving 10 mg of α-solanine (Sigma-Aldrich, Poznań, Germany) in 10 mL of methanol and 5 mg of α-chaconine (Sigma-Aldrich, Germany) in 5 mL of methanol. A calibration curve was prepared in the range from 1 to 50 μg·mL^−1^ for both analytes. Twenty microliters were injected into the column.

### 2.7. Apparatus and Conditions of the Glycoalkaloids Separation

To determine the content of α-solanine and α-chaconine, a high-pressure liquid chromatography HPLC (Prominence-i LC-2030C Plus) was used, made by Shimadzu Corporation (Kyoto, Japan), equipped with LC-2030 UV detector, Supelcosil LC-18 (25 cm × 4.6 mm, 5 µm) analytical column, (Supelco Inc., Bellefonte, PA, USA) and a computer system monitoring the chromatograph (Shimadzu LabSolutions, Darmstadt, Germany). As a mobile phase, 0.1 M KH2PO_4_ and acetonitrile (70:30 *v*/*v*) was used. The separation was performed at 70 °C with a 1 mL⋅min^−1^ flow rate, applying the light wavelength of 200 nm and injection volume of 20 μL.

### 2.8. Statistical Analysis

The results obtained in the experiment were subjected to statistical calculations of the Statistica v. 13.1 software StatSoft: Tulsa, OK, USA [31]. ANOVA/MANOVA analysis of variance of the data. The Duncan test was performed for comparing the means, and homogeneous groups were determined to prove the significance of the observed differences, using multiple comparisons, and the standard deviations (±SDs) were estimated. All experiments were performed in two technological replications, and the present results show the average values obtained in this investigation.

## 3. Results and Discussion

### 3.1. Raw Material Characteristics

The two analyzed varieties of potatoes with colored flesh, namely Mulberry Beauty (MB) with red flesh and Double Fun (DF) with purple flesh, slightly differed only in terms of dry matter and the content of starch and reducing sugars (Table 1). The analyzed samples contained 21.75–22.24 g·100^−1^ g dry weight (d.w.), 16.08–16.66 g·100^−1^ g starch and 0.18–0.22 g·100^−1^ g reducing sugars, and for this reason, they were a suitable raw material for the production of fried snacks [32]. However, they varied to a large extent in content of α-solanine, α-chaconine, and the sum of these compounds (TGA), as well as the ratio of both these of glycoalkaloids forms. Mulberry Beauty potatoes contained more of both forms of glycoalkaloids. The share of α-chaconine in potatoes of the Double Fun variety was higher (2.8) than in the Mulberry Beauty variety (2.4). Urban et.al. [12] determined the content of glycoalkaloids in fourteen new potato cultivars with purple and red flesh in comparison with yellow- and white-fleshed control potatoes; the TGA levels in tubers’ flesh ranged from 33.69 to 167.77 mg·kg^−1^ fresh matter (FM), and the ratio of α-chaconine to α-solanine ranged from 1.18 to 3.78. Friedman and Levin [19] report that the TGA content of tubers of potatoes with different color flesh ranges from 8.0 to 63.1 mg·kg^−1^ FW.

The authors Ieri et al. [3], Urban et al. [12], and Friedman and Levin [19] also stated that a decisive influence on TGA content had the cultivar genotype, not flesh color. Jansen and Flamme [11] showed the presence of total glycoalkaloids in raw potatoes of the red-fleshed Red Cardinal (2.86 mg·100 g^−1^ FW) and in the blue-fleshed varieties Blaue Zimmerli and Violettfleischige 2.21 mg·100 g^−1^ and 7.5 mg·100 g^−1^ FW, respectively.

In the conducted research, the TGA content in potatoes of the MB variety on average stated about 6.04 mg·100 g^−1^ FM, more than found in the tubers of the Double Fun variety, i.e., 2.34 mg·100 g^−1^ FM; however, both of them were within the range of the permitted quantity for health reasons, which should not be higher than 12 mg·100 g^−1^ of raw tubers [10]. These amounts were lower than those obtained by other authors, such as Urban et al. [12], and analyzing the content of glycoalkaloids in potatoes of varieties with colored flesh, i.e., mostly in the range from 4.10 to 11.0 mg·100 g^−1^ FM. Lachman et al. [33] also found that the TGA levels in red-fleshed potato varieties were higher compared to blue-fleshed varieties.

The tested samples of tubers of both potato varieties differed in terms of their total anthocyanin content. Potatoes of the DF variety with purple flesh contained twice as much of these compounds, i.e., on average, 50.38 mg·100 g^−1^ FW (Table 1). Lachman et al. [33] reports that potatoes with red flesh contained an average of 231.9 mg·100 g^−1^ DM of anthocyanins, while those with purple flesh contained 146.8 mg·100 g^−1^ DM.

The conducted research assumed a beneficial effect of the potato blanching process on reducing the content of glycoalkaloids, as confirmed by the authors Tian et al. [7], Singh et al. [17], Lachman et al. [33], Omayio et al. [34], and Nie et al. [35], who noted that microwaving and baking processes reduced the amount of glycoalkaloids in unpeeled potatoes by 45% and 51%, respectively, compared to the raw material. The data in Table 2 show that immersing blanched potato slices in 1% solutions of five organic acids contributed to reducing the amount of glycoalkaloids in the resulting experimental droughts, compared to the raw material—dried, unpeeled tubers (Table 1). The decrease in the amount of glycoalkaloids depended on both the potato variety and the type of organic acid in which the blanched potato slices were soaked. The TGA content in experimental grits decreased in relation to the raw material by 49–79% when MB variety potatoes were used and by 19–57% when DF variety potatoes were used. Experimental potato grits made from potatoes of the MB variety contained, apart from samples soaked in citric and tartaric acids, more glycoalkaloids than grits obtained from tubers of the DF variety.

The lowest TGA content in experimental grits was found in samples obtained from tubers of the MB variety treated with tartaric and citric acid, and from potatoes of the DF variety treated with ascorbic acid. With regard to the raw material, in experimental grits, especially those obtained from potatoes of the DF variety, the content of α-solanine decreased under the influence of blanching and immersion in acid solutions to a greater extent than α-chaconine, with the exception of samples immersed in malic acid (Table 1 and Table 2). As a result of the effect of technological factors, including the type of organic acid used, the ratio of the amount of α-chaconine to α-solanine changed. In most experimental grits from tubers of the DF variety, this parameter increased to 3.6–3.9, to the greatest extent after the use of citric and tartaric acids. In samples of grits from (MB) tubers treated with citric and tartaric acids was recorded an increase in the proportion of α-chaconine in TGA to 2.5–2.9 and in the control sample to 3.1.

When analyzing the obtained results, it can be concluded that immersing blanched potato slices in solutions of appropriately selected low-concentration organic acids may contribute to a significant reduction in the amount of toxic glycoalkaloids in products obtained from varieties of potatoes of colored flesh.

Research by various authors, such as Ieri et al. [3], Tian et al. [7], D’Amelia et al. [13], Rytel et al. [15], Lachman et al. [33], Nie et al. [35], and Rytel et al. [36], shows that glycoalkaloids contained in potatoes can be largely removed during their processing, as a result of rinsing, but also as a result of the degradation of these compounds at raised temperatures, e.g., during frying or baking processes. From the other side, information in the literature on the influence of various organic acids on changes in the glycoalkaloid content in color-fleshed potatoes during processing and in ready products is scarce.

### 3.2. Influence of Different Factors in the Production of Pellet Snacks on the Content of Glycoalkaloids in Ready Products

The potato grits obtained during the experimental research were, next to potato starch, the basic ingredient of extruded pellets, which in turn were a semi-product for obtaining expanded, crispy snacks of the third generation. A highly significant effect of the factors used in the research on the content of glycoalkaloids in ready-made snacks was found (Table 3). Products obtained with grits from DF variety tubers contained more α-chaconine and TGA than those made with grits from MB variety potatoes; however, regardless of the origin of the grits, the amount of glycoalkaloids in the snacks remained at a low level, α-solanine did not exceed 0.47 mg·100 g^−1^, and α-chaconine 0.79 mg·100 g^−1^, so there was a total of 1.26 mg·100 g^−1^ TGA.

The statistical analysis of the one-way ANOVA data (Table 3) and interactions (Figure 2A; Appendix A) showed that fried snacks obtained from pellets based on grits from tubers of both potato varieties contained 62–64% less glycoalkaloids than those expanded under the influence of microwaves, and, accordingly, the content of glycoalkaloids in fried snacks from MB variety potatoes reached 0.38 mg·100 g^−1^ TGA, and from DF variety potato tubers, 0.52 mg·100 g^−1^ TGA, while in samples expanded under the influence of microwaves, these values were 1.27 and 1.24 mg·100 g^−1^ TGA, respectively. Regardless of the snack expansion method, they contained more α-chaconine than α-solanine. In snacks made from MB potatoes, the proportion of α-chaconine to α-solanine was 1.59, and in products made from DF potatoes, it was 1.82 (Figure 2; Appendix A).

These differences between fried and microwaved snacks were probably related to the presence of fat in fried snacks and a lower share of the dry matter of potato tubers and other ingredients of the crisp recipe in the weight of these products. Fried potato or snacks of the third generation usually contain about 20–30% fat [32,37].

Organic acids used in the process of obtaining experimental potato grits had a different impact on the content of glycoalkaloids in the snacks obtained with them. The use of tartaric or ascorbic acid contributed to a higher content of both forms of glycoalkaloids in the snacks: on average, they contained 0.35 mg·100 g^−1^ α-solanine, 0.64 mg·100 g^−1^ α-chaconine, and a total of TGA 0.99 mg·100 g^−1^ (Table 3). On the other side, the use of malic acid in the process of obtaining snacks, as well as citric acid, resulted in a reduction in the glycoalkaloid content. The products of the control sample, in which no organic acid was used—only water—were similar in terms of glycoalkaloid content to the remaining crisp samples, except for the samples obtained with tartaric acid.

Regardless of the pellet expansion method, there was a significant impact of the potato variety and the type of organic acid used on the glycoalkaloid content in ready-to-eat snacks (Figure 2B; Appendix A).

Tartaric acid contributed to a reduction in the content of α-chaconine in products based on grits from tubers of both varieties, while ascorbic acid caused a greater increase in the content of α-chaconine in snacks made from potatoes of the DF variety. Compared to the control sample, the use of organic acids, apart from malic acid, contributed to a slight increase in the α-solanine content in the ready products, especially those made with grits from DF variety tubers. The content of total glycoalkaloids in snacks made with organic acids differed only slightly from the control sample when the raw material was MB variety potatoes. However, it was shown that the use of organic acids in the production of grits from potatoes of the DF variety with purple flesh, apart from malic acid, resulted in an increase in the TGA content in the analyzed products, compared to the control sample.

As shown from the data present in Figure 2C and Appendix A, the content of glycoalkaloids in the analyzed snacks depended more on the method of expanding the pellets than on the type of organic acid used for their production. Fried products from the control sample contained over 3.3 times less TGA than microwaved snacks. Fried products made with organic acids contained 2.2 (malic acid) to 3.5 (lactic acid) times less TGA compared to microwave samples. The greatest differences in the content of individual forms of glycoalkaloids and their sum between the fried and microwaved snacks samples occurred in the samples in which lactic acid was used. The differences in the content of α-solanine, α-chaconine and TGA in these snack products were 4.2, 3.3, and 3.5 times lower, respectively, in fried products than in microwaved products. The smallest differences were observed in snack samples prepared with malic acid, which contained 2.4, 2.1, and 2.2 times less glycoalkaloids in the fried samples than in the samples heated in a microwave oven, respectively. Thus, malic acid turned out to be the most preferred organic acid in terms of the glycoalkaloids content in snacks made from potatoes of colored flesh varieties, but it was particularly beneficial in the production of microwaved snacks. The available literature lacks information on the impact of the use of organic acids in the production of fried products from potatoes with red or purple flesh. Research involving acids primarily examines the specific color of the resulting products and their texture [38]. In contrast, some authors, such as Huang et al. [39] and Negoiță et al. [40], used the addition of blanching acids, i.e., citric, acetic, and ascorbic, during the production process of fried potato snacks in order to reduce the amount of acrylamide.

By comparing the content of glycoalkaloids in the tested fried and microwaved snacks with their content in the raw material, i.e., in the experimental grits, at the same level of dry matter, it was found that, in the fried crisps made with grits from Mulberry Beauty tubers, from 2.95 to 5.86% α-solanine remained, from 3.17 to 4.68% α-chaconine, and from 3.31 to 4.81% TGA; and in crisps containing Double Fun potato grits, from 9.32 to 15.9% α-solanine, from 5.78 to 8.98% α-chaconine, and from 6.14 to 9.27% TGA. Higher glycoalkaloid residues were found in microwaved crisps than in fried snacks. Depending on the type of acid used, glycoalkaloid residues in potato snacks ranged from 11.5 to 15.0% TGA when experimental grits from Mulberry Beauty potato tubers were used and from 22.6 to 23.7% TGA when it was Double Fun potato grits. However, more α-solanine than α-chaconine remained in the snacks.

Research conducted by Rytel et al. [15] showed that the French fries they obtained contained 3%, crisps 16%, and dried potatoes 17% of the initial amount of total glycoalkaloids (TGA) contained in the raw material, and the reduction in the content of α-chaconine and α-solanine was at a similar level.

### 3.3. Influence of Different Factors in the Production of French Fries on the Content of Glycoalkaloids in Ready Products

The content of both forms of glycoalkaloids in French fries obtained during the research from potatoes with colored flesh depended primarily on the potato variety and the method of preparing the fries for consumption, and only to a small extent on the type of organic acid used at the stage of blanching the potato strips (Table 4). The interactions of the factors used had a significant impact on the content of glycoalkaloids in ready-made French fries (Figure 3A–C; Appendix A).

French fries obtained from tubers of the purple-colored DF variety contained, on average, 2 times less α-solanine compared to French fries prepared from tubers of the MB variety with red flesh, while the differences between French fries from potatoes of these varieties in terms of α-chaconine and TGA content were minor. A more advantageous method of preparing French fries for consumption in terms of the glycoalkaloid content in the final product was baking a frozen semi-finished product. This concerned the content of α-solanine to a greater extent. Baked French fries contained on average approximately 3 times less α-solanine than fried French fries. Regardless of the type of acid used, the content of α-solanine in the tested French fries decreased compared to the control sample (immersion in water), while the content of α-chaconine and TGA increased, especially after the use of ascorbic acid (Table 4).

A more pronounced influence of the method of preparing French fries on the content of glycoalkaloids was found in products obtained from potatoes of the MB variety (Figure 3A; Appendix A). French fries fried in the second stage, obtained from tubers of this variety, contained more of both forms of glycoalkaloids than baked ones. However, such relationship was not found in French fries obtained from potatoes of the DF variety, with purple flesh. The process of soaking potato strips in solutions of various organic acids turned out to be particularly beneficial in terms of glycoalkaloid content for French fries obtained from potatoes of the MB variety. These samples contained fewer glycoalkaloids as compared to the control products, regardless of the type of organic acid used (except ascorbic acid). Potatoes of the DF variety reacted in the opposite way to the use of organic acids. The lactic and tartaric acids were the most beneficial acids in the production of French fries from MB variety tubers, while the citric acid was found to be the best in the production of French fries from DF variety potatoes (Figure 3B; Appendix A). Ascorbic acid turned out to be unfavorable acid in the production of French fries when the MB variety was used, and ascorbic and tartaric acids when the DF variety was used. The method of preparing the French fries to consumption had a significant influence on the content of both forms of glycoalkaloids and their sum (TGA). The data in Figure 3C and Appendix A show that, regardless of the method of preparing French fries, i.e., frying or baking, the α-solanine content in the ready-to-eat French fries was at a low level: in fried samples, it was found from 0.13 to 0.34 mg·100 g^−1^ of product, and in baked samples, from 0.05 to 0.10 mg·100 g^−1^ of product. The use of organic acids in the production of deep-fried French fries contributed to a reduction in the α-solanine content compared to the control sample, while in the samples of baked French fries, there was no effect of the type of acid on the content of this form of glycoalkaloid.

The use of malic and tartaric acids turned out to be particularly beneficial. In the French fries obtained in the study, regardless of the method of their preparation for consumption, the α-chaconine content predominated. Fried samples of ready products contained from 1.39 to 1.61 mg·100 g^−1^ of this alkaloid, while baked samples contained from 0.83 to 1.75 mg·100 g^−1^ (Figure 3C; Appendix A). The content of total glycoalkaloids ranged from 1.50 to 1.94 mg·100 g^−1^ of fried French fries and from 0.89 to 1.83 mg·100 g^−1^ of baked ones. In fried French fries, the use of lactic, malic, and tartaric acids contributed to a greater reduction in the content of α-chaconine and TGA than when other acids were used.

Baked products from the control sample and samples in which citric and lactic acid were used contained approximately 37–53% less α-chaconine and approximately 32–51% less total glycoalkaloids as compared to French fries obtained with ascorbic acid. Thus, in the production of baked French fries, the use of citric and lactic acids turned out to be beneficial due to the content of glycoalkaloids, and in the production of deep-fried fries, the use of lactic, malic, and tartaric acids turned out to be favorable. In a study conducted by Liu et al. [41], soaking potato strips in clean water for 8 h had a similar effect on reducing the glycoalkaloid content in raw potatoes as immersing them in acetic acid solutions for the same time. As a result, less than 8.6% of α-solanine and α-chaconine remained in the raw potato samples, both in the control sample and after acid application, especially after longer soaking times, i.e., after 12 and 24 h. In our experiment, a significant effect in reducing the content of glycoalkaloids in products from tubers of varieties with colored flesh was achieved after just 5 min of soaking in 1% solutions of the five acids used. The analysis of the glycoalkaloid content in ready-made French fries in relation to the amount of glycoalkaloids present in raw, unpeeled tubers intended for the production of French fries, carried out at the same DM content, showed that α-solanine residues in fried French fries ranged from 1.65 to 14.6%, α-chaconine from 11.7 to 42.7% and TGA from 8.84 to 34.9%, with larger amounts of glycoalkaloids coming from tubers of the Double Fun variety. Baked fries contained from 1.2 to 9.71% of α-solanine derived from the raw material, from 6.78 to 12.5% of α-chaconine, and from 5.15 to 44.2% of TGA, with higher values for fries obtained from Double Fun potatoes. Liu et al. [41] obtained an over 90% reduction in the content of glycoalkaloids in fried French fries in relation to raw tubers by soaking potato pieces in acetic acid solutions for a period of 1 to 8 h. These authors examined potatoes of a light-fleshed variety and found that the reduction in the content of glycoalkaloids in fries was mainly influenced by the soaking time and not the concentration of the solution of this acid. It should also be noted that the heat treatment of the potatoes themselves can have an impact on reducing the amount of TGA in the ready products. D’Amelia et al. [13] report that cooking potatoes had a reduction effect on the amounts for α-solanine and α-chaconine of about 80% and 65%, respectively, while the frying process reduced the amount of total TGA by about 90% on average. According to Nie et al. [35], the frying process of chips had an effect of reducing the amount of TGA by about 94% compared to tubers. Authors D’Amelia et al. [13] also found that the use of microwave and oven heat treatments had an effect on reducing the amount of glycoalkaloids (α-solanine and α-chaconine), but less than the frying process.

## 4. Conclusions

Processing potatoes containing anthocyanins requires maintaining the structure and color of these phenolic compounds and their health-promoting effects. Processing tubers with skin increases the amounts of anthocyanins in the products obtained from them, but it is possible to introduce other compounds found in potato skin into the finished products, including glycoalkaloids, which are toxic in larger amounts.

The research showed a varied impact of the use of 1% solutions of ascorbic, citric, lactic, malic, and tartaric acids in the production of dried potato groats and third-generation snacks and French fries obtained from them on the content of α-solanine and α-chaconine. The use of tartaric acid in the production of third-generation snacks contributed to an increase in the content of TGA and α-chaconine, regardless of the potato variety. Malic acid turned out to be the most beneficial in their production, regardless of the potato variety and the method of expanding the pellets. Due to the content of glycoalkaloids, a more favorable method of expanding the pellets was deep-frying in hot oil. Third-generation fried snacks contained, on average, three times less glycoalkaloids than those from the microwave, which may be related to the higher dry matter content of potato snacks that came from the raw material.

In the production of French fries, the use of organic acids had little or no effect on the glycoalkaloid content in the finished products and depended on the potato variety. In this respect, it was more advantageous to use potatoes of the MB variety with red flesh and the use of lactic acid and baking fried and frozen semi-finished products rather than deep-frying them in oil. Regardless of the potato variety, the process of baking fried French fries turned out to be more beneficial than frying them due to the three times lower content of glycoalkaloids in the finished products. This was probably related to the greater degradation of glycoalkaloids in a longer process and higher temperature. Studies have shown a greater impact of technological treatments and organic acids on reducing the content of less toxic α-solanine in finished products. With regard to the raw material, 3.31–9.27% TGA remained in third-generation fried snacks, from 11.5 to 23.7% TGA in microwaved snacks, from 8.84 to 34.9% TGA in fried French fries, and from 5.15 to 44.2% TGA in baked fries.

## Figures and Tables

**Figure 1 foods-13-01712-f001:**
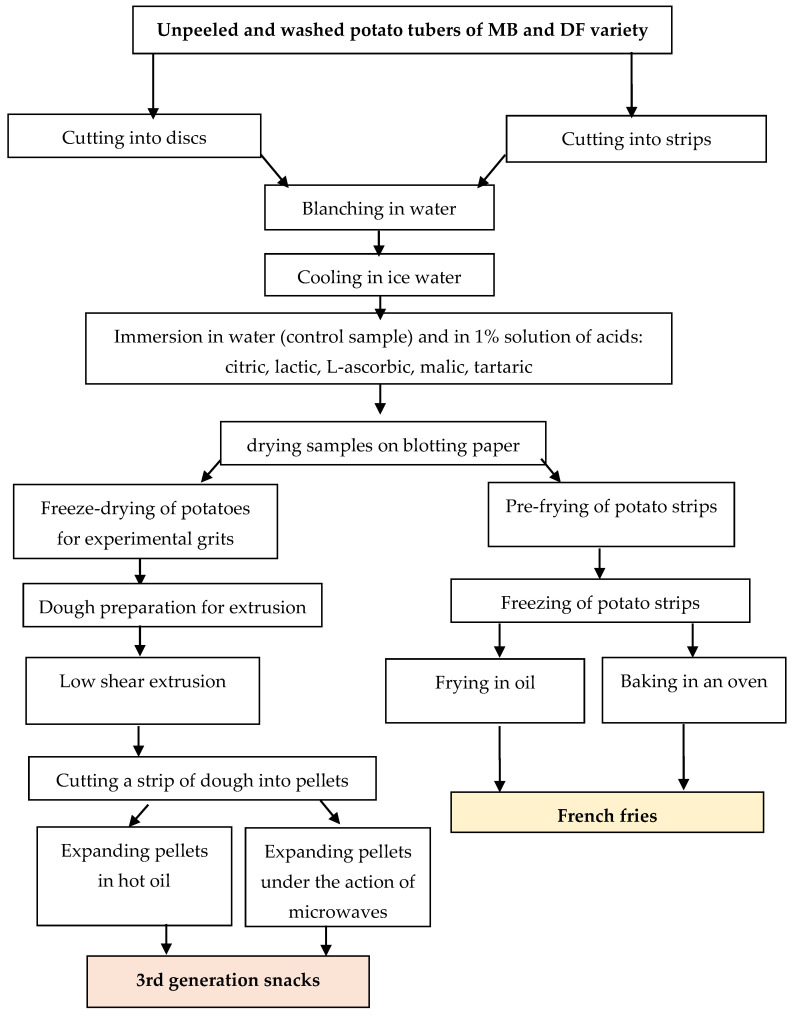
Scheme of laboratory preparation of third-generation snacks and French fries from potatoes of 2 varieties with colored flesh.

**Figure 2 foods-13-01712-f002:**
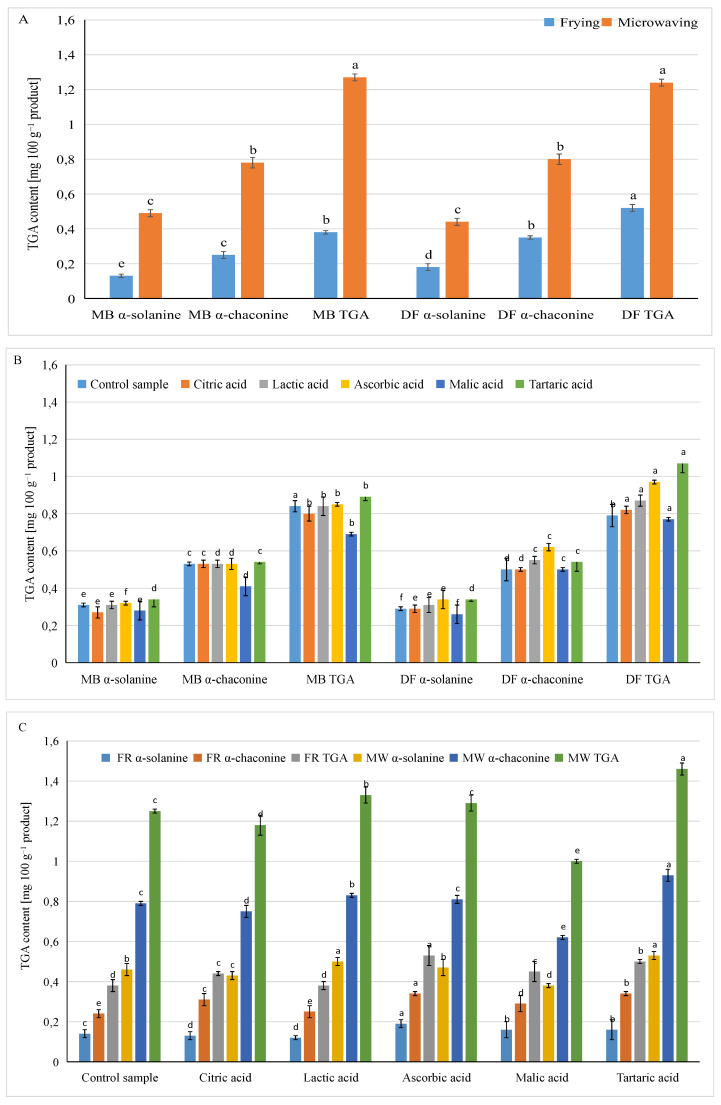
The influence of (**A**) potato variety and expanding method, (**B**) potato variety and acid type, and (**C**) acid type and expansion method on the glycoalkaloids content in snacks. MB—Mulberry Beauty; DF—Double Fun; TGAs—total glycoalkaloids; FR—frying; MW—microwave. Values are represented as mean (±SD) standard deviation (*n* = 6); ^a–f^—indicate significant differences. (**A**) Potato variety and expanding method, (**B**) potato variety and acid type, and (**C**) acid type and expansion method (Duncan’s test, *p* ≤ 0.05).

**Figure 3 foods-13-01712-f003:**
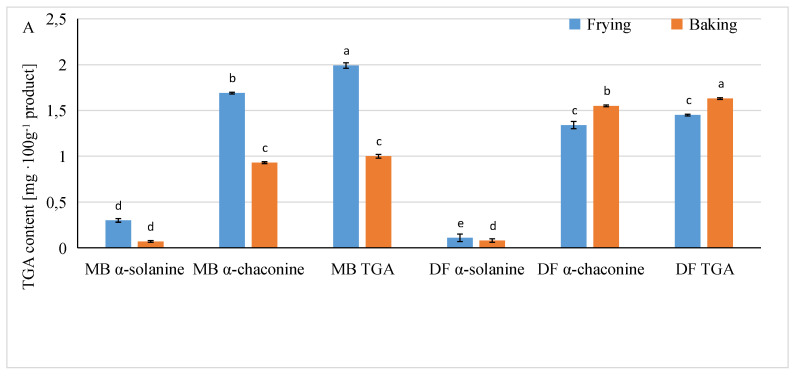
The influence of the (**A**) preparing method and potato variety, (**B**) acid type and potato variety, and (**C**) preparing method and acid type on the glycoalkaloids content in French fries made from colored potatoes of two varieties. MB—Mulberry Beauty; DF—Double Fun; FR—frying. Values are represented as mean (± SD) standard deviation, (*n* = 6). ^a–f^ Significant differences. (**A**) Preparing method and potato variety, (**B**) acid type and potato variety, and (**C**) preparing method and acid type (Duncan’s test, *p* ≤ 0.05).

**Table 1 foods-13-01712-t001:** Characteristics of studied potatoes of two varieties.

Potato Variety	Compound	Raw Unpeeled Potatoes	Dried Unpeeled Potatoes
	Dry matter [g·100 g^−1^]	22.24 ± 0.23 ^A^	92.88 ± 0.14 ^A^
	Starch [g·100 g^−1^]	16.66 ± 0.11 ^A^	69.58 ± 0.07 ^A^
	Reducing sugars [g·100 g^−1^]	0.22± 0.08 ^A^	0.92 ± 0.05 ^A^
	α-solanine [mg·100 g^−1^]	1.78 ± 0.02 ^A^	7.42 ± 0.01 ^A^
(MB)	α-chaconine [mg·100 g^−1^]	4.33 ± 0.04 ^A^	18.09 ± 0.02 ^A^
	TGA [mg·100 g^−1^]	6.04 ± 0.11 ^A^	25.21 ± 0.13 ^A^
	α-chaconine/α-solanine	2.4	2.4
	A sum of anthocyanins [mg·100 g^−1^]	24.95 ± 0.13 ^B^	104.2 ± 0.15 ^B^
	Dry matter [g·100 g^−1^]	21.75 ± 0.19 ^B^	92.68 ± 0.25 ^A^
	Starch [g·100 g^−1^]	16.06 ± 0.15 ^A^	68.43 ± 0.03 ^B^
	Reducing sugars [g·100 g^−1^]	0.18 ± 0.02 ^A^	0.77 ± 0.01 ^A^
	α-solanine [mg·100 g^−1^]	0.61 ± 0.03 ^B^	2.60 ± 0.02 ^B^
(DF)	α-chaconine [mg·100 g^−1^]	1.73 ± 0.01 ^B^	7.36 ± 0.02 ^B^
	TGA [mg·100 g^−1^]	2.34 ± 0.03 ^B^	9.96 ± 0.02 ^B^
	α-chaconine/α-solanine	2.8	2.8
	A sum of anthocyanins [mg·100 g^−1^]	50.38 ± 0.15 ^A^	214.7 ± 0.11 ^A^

^(A,B)^—the same capital letters within the same column were not significantly different at (*p* < 0.05); according to Duncan’s least significant difference test; mean values (*n* = 6); ±SD—standard deviation; MB—Mulberry Beauty; DF—Double Fun.

**Table 2 foods-13-01712-t002:** Characteristics of experimental potato grits depending on the type of organic acid used.

Potato Variety	Compound			Experimental Dried Potato			
		Water/Control	Citric Acid	Lactic Acid	Ascorbic Acid	Malic Acid	Tartaric Acid
	Dry matter [g·100 g^−1^]	92.21 ± 0.23 ^B^	93.05 ± 0.17 ^A^	93.10 ± 0.16 ^A^	92.75 ± 0.20 ^A^	93.08 ± 0.14 ^A^	92.99 ± 0.19 ^A^
	α-solanine [mg·100 g^−1^]	2.32 ± 0.03 b^A^	1.50 ± 0.06 ^cB^	3.82 ± 0.01 ^aA^	2.66 ± 0.02 ^bA^	3.96 ± 0.03 ^aA^	1.52 ± 0.02 ^cB^
(MB)	α-chaconine [mg·100 g^−1^]	7.46 ± 0.02 b^A^	4.44 ± 0.02 ^cdB^	9.12 ± 0.03 ^aA^	5.25 ± 0.03 ^cA^	8.20 ± 0.01 ^abA^	3.82 ± 0.03 ^dB^
	TGA [mg·100 g^−1^]	9.78 ± 0.08 b^A^	5.94 ± 0.01 ^dB^	12.94 ± 0.02 ^aA^	7.91 ± 0.02 ^cA^	12.16 ± 0.02 ^aA^	5.34 ± 0.01 ^dB^
	α-chaconine/α-solanine	3.1	2.9	2.4	2.0	2.1	2.5
	Dry matter [g·100 g^−1^]	93.39 ± 0.18 ^A^	92.91 ± 0.17 ^B^	93.54 ± 0.15 ^A^	92.77 ± 0.19 ^A^	92.30 ± 0.21 ^B^	93.01 ± 0.14 ^A^
	α-solanine [mg·100 g^−1^]	1.54 ± 0.01 ^abB^	1.58 ± 0.05 ^abA^	1.29 ± 0.06 ^cB^	0.92 ± 0.02 ^cB^	1.70 ± 0.01 ^aB^	1.66 ± 0.04 ^aA^
(DF)	α-chaconine [mg·100 g^−1^]	5.51 ± 0.03 ^abB^	6.10 ± 0.06 ^aA^	4.69 ± 0.08 ^abB^	3.37 ± 0.02 ^bB^	3.88 ± 0.02 ^bB^	6.44 ± 0.03 ^aA^
	TGA [mg·100 g^−1^]	7.04 ± 0.02 ^abB^	7.68 ± 0.04 ^abA^	5.98 ± 0.07 ^bcB^	4.29 ± 0.02 ^cB^	5.58 ± 0.01 ^bcB^	8.10 ± 0.03 ^aA^
	α-chaconine/α-solanine	3.6	3.9	3.6	3.7	2.3	3.9

^(a–d)^ The same lower letters within the same rows were not significantly different at *p* < 0.05; ^(A,B)^ the same capital letters within the same column were not significantly different at (*p* < 0.05); according to Duncan’s least significant difference test; mean values (*n* = 6); ±SD—standard deviation; MB—Mulberry Beauty; DF—Double Fun.

**Table 3 foods-13-01712-t003:** Results of ANOVA and LSD multiple range tests of the influence of the pellet expansion method, the type of organic acid used, and the potato variety on the content of glycoalkaloids in the obtained third-generation snacks.

Factor	α-Solanine	α-Chaconine [mg·100 g^−1^]	TGA
ANOVA Test			
Potato variety	NS	***	***
Expanding method	***	***	***
Acid type	***	***	***
Potato variety x acid type	***	***	***
Expanding method x acid type	***	***	***
Potato variety x expanding method	***	***	***
LSD multiple range test			
Expanding method			
Frying	0.15 b	0.30 b	0.45 b
Microwaving	0.47 a	0.79 a	1.26 a
Type of acid			
Control	0.30 cd	0.52 bc	0.82 c
Citric acid	0.29 c	0.53 b	0.81 cd
Lactic acid	0.32 ab	0.54 b	0.86 bc
Ascorbic acid	0.33 ab	0.58 ab	0.91 ab
Malic acid	0.27 c	0.46 c	0.73 d
Tartaric acid	0.35 a	0.64 a	0.99 a
Potato variety			
Mulberry Beauty	0.31 a	0.51 b	0.82 b
Double Fun	0.31 a	0.57 a	0.88 a

NS = not significant at *p* > 0.05; *** significant at *p* < 0.001, respectively; (*n* = 6), a–d—values followed by the same letter, within the same column, were not significantly different (*p* > 0.05), according to Duncan’s least significant difference test.

**Table 4 foods-13-01712-t004:** Results of ANOVA and LSD multiple range tests of the influence of French fries’ preparing method, the type of organic acid used, and the potato variety on the content of glycoalkaloids in ready products.

Factor	α-Solanine	α-Chaconine [mg·100 g^−1^]	TGA
ANOVA Test			
Potato variety	***	***	***
Preparing method	***	***	***
Acid type	*	*	NS
Potato variety x preparing method	***	***	***
Preparing method x acid type	***	**	**
Potato variety x acid type	***	***	***
LSD multiple range test			
Preparing method			
Frying	0.20 a	1.51 a	1.71 a
Baking	0.08 b	1.24 b	1.32 b
Type of acid			
Control	0.20 a	1.23 b	1.43 a
Citric acid	0.14 ab	1.22 b	1.36 a
Lactic acid	0.17 ab	1.26 b	1.43 a
Ascorbic acid	0.13 ab	1.67 a	1.80 a
Malic acid	0.09 b	1.40 ab	1.50 a
Tartaric acid	0.10 ab	1.49 ab	1.59 a
Potato variety			
Mulberry Beauty	0.18 a	1.31 b	1.49 a
Double Fun	0.09 b	1.45 a	1.54 b

NS = not significant at *p* > 0.05; *, **, and *** significant at *p* < 0.05, 0.01, and 0.001, respectively (*n* = 6); a,b—values followed by the same letter, within the same column, were not significantly different (*p* > 0.05), according to Duncan’s least significant difference test.

## Data Availability

The original contributions presented in the study are included in the article/Appendix A, further inquiries can be directed to the corresponding author.

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
