# Peer review of "Assessment of the Content of Glycoalkaloids in Potato Snacks Made from Colored Potatoes, Resulting from the Action of Organic Acids and Thermal Processing"

_foods, 2024, doi:10.3390/foods13111712_

Round 1

Reviewer 1 Report

Comments and Suggestions for Authors

The manuscript entitled “Study on the glycoalkaloids content in coloured-flesh potato products produced using various organic acids and final treatments” investigates how the addition of organic acids affects the content of toxic glycoalkaloids in different potato varieties and processing methods. The topic is interesting and there are a few minor issues found that could be addressed to improve its quality.

1.       Due to the significant number of experiments performed, include in the manuscript experimental flow chart or table, outlining the conducted experiments.

2.       Line 111-112:  Briefly describe the method for the anthocyanin content determination.

3.       Table 1 is difficult to understand, make it more presentable. In its current form, it's challenging to follow, primarily because the numbers are too close together and the columns are not neatly divided.

4.       Considering that the data presented in Tables 2 and 3 are already given in the text and presented in corresponding graphs, I suggest relocating Tables 2 and 3 to the Supplementary Information for better clarity.

5.       Given that color is one of the key attributes of these potato varieties, how did the addition of organic acids affect the color of potatoes and the resulting products?

Author Response

Responses to Reviewer 1 comments

The authors are very grateful to the Reviewer for his time and valuable comments and guidance on the manuscript, which increased its merit and were very helpful. In the responses to the Reviewer and in the manuscript, the amendments has been indicated in blue font.

Reviewer 1

The manuscript entitled “Study on the glycoalkaloids content in coloured-flesh potato products produced using various organic acids and final treatments” investigates how the addition of organic acids affects the content of toxic glycoalkaloids in different potato varieties and processing methods. The topic is interesting and there are a few minor issues found that could be addressed to improve its quality.

Reviewer's comment 1: Due to the significant number of experiments performed, include in the manuscript experimental flow chart or table, outlining the conducted experiments.

Response 1: According the Reviewer suggestion the scheme (Figure 1. Scheme of laboratory preparation of third-generation snacks and French fries from potatoes of 2 varieties with colored flesh) has been added in section 2.3., on page 4 in lines from 158 to 198. Starting from line 155 to 157, the following information was added: “The scheme for preparing experimental potato grits, intermediate products, ready-made snacks and French fries is shown in Figure 1”

Reviewer's comment 2: Line 111-112:  Briefly describe the method for the anthocyanin content determination

Response 2: According the Reviewer suggestion has been added the following information in section 2.2. on page.3 in lines from 124 to 142.

The anthocyanin content was determined using the HPLC method as described below. Freeze-dried potato samples were extracted with 70% aqueous acetone (acidified with 0.1% acetic acid). The homogenized mixture was left for 2 hours at room temperature, and then the acetone-water mixture was separated with chloroform to remove lipophilic compounds. The acetone-water fraction was collected and the remaining acetone was evaporated on a Büchi rotary evaporator. The obtained aqueous extract was brought to a known volume with distilled water and stored at 20°C until analysis. Samples filtered through 0.2 µm filters were used for HPLC analysis of anthocyanins. Anthocyanins were determined using a Dionex HPLC system (USA) equipped with an Ultimate 3000 model diode detector, an LPG-3400A quaternary pump, an EWPS-3000SI automatic sampler and a TCC-3000SD thermostabilized column chamber, controlled by Chromeleon v.6.8 software. An Atlantic T3 reversed-phase column (250 mm x 4.6 i.d., 5 μm) (Waters, Ireland) and an Atlantis T3 guard column (20 x 4.6 i.d., 5 μm) (Waters, Ireland) were used. The following solvents were used as the mobile phase: A/ 4.5% formic acid and B/ acetonitrile. The following elution conditions were used: 0–1 min, 5% B isocratic; 1–6 min, linear gradient from 5–10% B; 6–26 min, linear gradient 10–20% B; 26–33 min, linear gradient from 20–100% B; and then the initial conditions. The flow rate was 1 ml·min-1 and the injection volume was 40 μL. The column was operated at 30°C. Anthocyanins were monitored at a wavelength of 520 nm. 

Reviewer's comment 3: Table 1 is difficult to understand, make it more presentable. In its current form, it's challenging to follow, primarily because the numbers are too close together and the columns are not neatly divided.

Response 3: According the Reviewer suggestion Table 1 has been divided into 2 separate tables (Table 1. Characteristics of studied potatoes of two varieties and Table 2. Characteristics of experimental potato grits depending on the type of organic acid used). Two tables has been appropriately described in the manuscript (text in blue font).  Table 1 has been placed on page 7 in lines from 298 to 301 and Table 2 – on page 8 in lines from 341 to 346.

Reviewer's comment 4:  Considering that the data presented in Tables 2 and 3 are already given in the text and presented in corresponding graphs, I suggest relocating Tables 2 and 3 to the Supplementary Information for better clarity

Response 4: According the Reviewer suggestion Tables 2 and 3 have been simplified. The experimental data contained therein have been transferred to the Supplementary  Materials as Table S1 and Table S2, respectively. Tables 2 and 3 have been given a new numbering, i.e.: 3 and 4, respectively. In Tables 3 and 4, the effects of the Anova and LSD multiple range tests have been left, illustrating significance of the influence of the factors used on the content of glycoalkaloids in snacks and French fries, respectively. The titles of these tables have also been changed: (Table 3. Results of ANOVA and LSD multiple range tests of the influence of the pellet expansion method, the type of organic acid used and the potato variety on the content of glycoalkaloids in the obtained third-generation snacks. Table 4. Results of ANOVA and LSD multiple range tests of the influence of French fries preparing method, the type of organic acid used and the potato variety on the content of glycoalkaloids in ready products). These tables has been appropriately described in the manuscript (text in blue font). Table 3 has been placed on page 10 in lines from 406 to 410 and Table 4 – on page 14 in lines from 538 to 543.

Reviewer's comment 5: Given that color is one of the key attributes of these potato varieties, how did the addition of organic acids affect the color of potatoes and the resulting products?

Response 5: The use of organic acids in the processing of colored potatoes containing various anthocyanins was aimed primarily at stabilizing the structure of these compounds and maintaining the traditional color of this raw materials used also in the products obtained from them. According the Reviewer suggestion has been added the following information on page 2 in lines from 73 to 81.

When processing potatoes with red and purple flesh, the colour of the finished products is also related to the transformation of anthocyanins that occur under the influence of high temperature [13,20,21]. In that conditions anthocyanins transform into colourless chalcones, which, when oxidized, can form high-molecular coloured compounds [22J. Due to the large number of factors, the mechanism of these transformations is not fully known. Anthocyanins have a high ability to attach numerous hydroxyl and methoxy groups in the flavylium cation ring. The degree of glycosylation and acylation of anthocyanidin with phenolic acids or organic acids improves the stability of anthocyanin pigments [23].

                                                                                            Yours sincerely

                                                                Agnieszka Tajner-Czopek and co-authors

Reviewer 2 Report

Comments and Suggestions for Authors

This manuscript aims to assess how various organic acids impact the toxic glycoalkaloid (TGA) content in snacks made from unpeeled coloured potatoes. Specifically, it investigates the effects on red-fleshed Mulberry Beauty (MB) and purple-fleshed Double Fun (DF) potato varieties, exploring different snack preparation methods such as frying, microwaving, and baking. The study aims to identify the most effective organic acids for minimizing TGA levels in the final snack products, thereby ensuring their safety and health benefits.

The manuscript is within the scope of the Journal Foods.

The manuscript needs some alterations.

Comments:

1.      The abstract does not provide any background information on glycoalkaloids or their potential health risks. This should be added.

2.      The abbreviation TGA should be introduced where first mentioned, not in line 66.

3.      Lines 39-64: This part is confusing, so it should be restructured.

4.      The novelty of this paper should be emphasized.

5.      Table 1 is crowded and should be revised.

6.      Line 404: Figure caption is missing.

7.      Figure numbers should be carefully checked.

8.      Line 491: selected 5 organic acids should be named here.

9.      In conclusion, some points are repeated or restated in different ways, which could be condensed to improve clarity and readability. The whole conclusion should be rewritten because it lacks an in-depth interpretation or discussion of the implications of the presented findings. The authors should provide clear recommendations or suggestions for future research based on the findings presented, as well as any limitations or potential biases in the study. 

Author Response

Responses to Reviewer 2 comments

The authors are very grateful to the Reviewer for his time and valuable comments and guidance on the manuscript, which increased its merit and were very helpful. In the responses to the Reviewer and in the manuscript, the amendments has been indicated in blue font.

Reviewer 2

This manuscript aims to assess how various organic acids impact the toxic glycoalkaloid (TGA) content in snacks made from unpeeled coloured potatoes. Specifically, it investigates the effects on red-fleshed Mulberry Beauty (MB) and purple-fleshed Double Fun (DF) potato varieties, exploring different snack preparation methods such as frying, microwaving, and baking. The study aims to identify the most effective organic acids for minimizing TGA levels in the final snack products, thereby ensuring their safety and health benefits. The manuscript is within the scope of the Journal Foods. The manuscript needs some alterations.

Reviewer's comment 1: The abstract does not provide any background information on glycoalkaloids or their potential health risks. This should be added.

Response 1: According the Reviewer suggestion the Abstract has been changed (on page 1 in lines from 13 to 25).

Abstract. Glycoalkaloids (TGA-Total glycoalkaloids), toxic secondary metabolites are found in potatoes (110-335 mg·kg-1 DW), mainly in the peel. Colourful, unpeeled potatoes are an innovative raw material for the production of snacks which are poorly tested in terms of their glycoalkaloid content. Third-generation snacks and French fries made from red-fleshed Mulberry Beauty (MB) and purple-fleshed Double Fun (DF) potatoes were produced with the use of 1% solutions of ascorbic, citric, lactic, malic and tartaric acids to stabilize the structure of anthocyanins in the raw material and maintain their colour in obtained products. The influence of the type of acid, thermal processes, like frying, microwaving and baking on the content of glycoalkaloids in ready-made products was examined. There were found only 0.45-1.26 mg ·100 g-1 of TGA in pellet snacks and 1.32-1.71 mg ·100 g-1 in French fries. Soaking blanched potatoes in organic acid solution reduced the α-chaconine content by 91-97% in snacks and by 57-93% in French fries in relation to the raw material, to the greatest extent after the use of malic acid and the DF variety. The effect of lactic and citric acid was also beneficial, especially in the production of baked French fries from MB potatoes.

Reviewer's comment 2: The abbreviation TGA should be introduced where first mentioned, not in line 66.

Response 2 : According the Reviewer suggestion the Abbreviation TGA has been explained in Abstract on page 1 in line 13.

Reviewer's comment 3: Lines 39-64: This part is confusing, so it should be restructured.

Response 3: According the Reviewer suggestion this part of Introduction has been simplified and placed on pages 1-2 in lines from 35 to 57.

Many studies are being carried out on increasing the amount or stabilizing the structure and activity of these ingredients in potatoes and their products, both for health reasons and sensory attractiveness. The importance of anthocyanins as active substances with antioxidant and free radical scavenging properties is emphasized, including their importance in inhibiting fat oxidation processes in fried potato products [2,9].

Potatoes of varieties with coloured flesh, like those with traditional flesh colour, contain glycoalkaloids produced in tubers as secondary metabolites. These compounds are represented in potatoes mainly by α-solanine and α-chaconine and exhibit toxic effects when their total amount reaches approximately 200 mg·kg-1 fresh weight FW, causing neurological disorders such as apathy, drowsiness, disorientation, and may even be fatal [10,11,12,13]. However, few studies also indicate the beneficial effects of small amounts of glycoalkaloids, including antipyretic and anti-inflammatory effects [2,10,14]. Research by these authors also shows that α-chaconine is a more toxic compound. Its content in potato tubers is 2-5 times higher than that of α-solanine. The total content of glycoalkaloids in potatoes ranges from 85 to 182 mg·kg-1 DM [2,10,11,14] and they are usually located at a distance of 1-1.5 mm from the outer part of the tuber [15]. The literature on the subject indicates that potatoes intended for food should not contain more than 12 mg·100 g-1 FW of total glycoalkaloids (TGA), as they are considered bitter and unacceptable to consumers. The result of research conducted by various research centres are potato varieties in which the accumulation of these compounds does not exceed 10 mg of glycoalkaloids in 100 g of tuber. Varieties with low TGA content also include, to a large extent, varieties with coloured flesh [10,16,17,18].

Reviewer's comment 4: The novelty of this paper should be emphasized.

Response 4: According the Reviewer suggestion this information has been added on page 2 in lines from 82 to 87.

The literature on the subject lacks information on the influence of various organic acids on the degree and direction of stabilization of the structure and colour of various acylated anthocyanins found in coloured potatoes, stable in a wide pH range. Potato varieties with coloured flesh are attracting the interest of an increasingly wide group of consumers in various countries around the world. Therefore, they are increasingly the subject of detailed research in terms of the possibility of their industrial processing.

Reviewer's comment 5: Table 1 is crowded and should be revised.

Response 5: According the Reviewer suggestion Table 1 has been divided into 2 separate tables (Table 1. Characteristics of studied potatoes of two varieties and Table 2. Characteristics of experimental potato grits depending on the type of organic acid used). Two tables has been appropriately described in the manuscript (text in blue font).  Table 1 has been placed on page 7 in lines from 298 to 301 and Table 2 – on page 8 in lines from 341 to 346.

Reviewer's comment 6:  Line 404: Figure caption is missing.

Response 6: According the Reviewer suggestion the title of missing Figure has been added on page 15 in lines from  546 to 554.

Reviewer's comment 7:  Figure numbers should be carefully checked.

Response 7: According the Reviewer suggestion the Figure numbers has been checked and corrected throughout the text (on pages from 10 to 16, lines from 412 to 599).

Reviewer's comment 8:  Line 491: selected 5 organic acids should be named here.

Response 8: According the Reviewer suggestion the information has been added and placed on page 16 in lines from 605 to 606.

Reviewer's comment 9: In conclusion, some points are repeated or restated in different ways, which could be condensed to improve clarity and readability. The whole conclusion should be rewritten because it lacks an in-depth interpretation or discussion of the implications of the presented findings. The authors should provide clear recommendations or suggestions for future research based on the findings presented, as well as any limitations or potential biases in the study. 

Response 9: According the Reviewer suggestion the Conclusions has been corrected and placed on pages 16-17 in lines from 600 to 628.

Processing potatoes containing anthocyanins requires maintaining the structure and colour of these phenolic compounds and their health-promoting effects. Processing tubers with skin increases the amount of anthocyanins in the products obtained from them, but it is possible to introduce other compounds found in potato skin into the finished products, including glycoalkaloids that are toxic in larger amounts.The research showed a varied impact of the use of 1% solutions of ascorbic, citric, lactic, malic and tartaric acids in the production of dried potato groats and third-generation snacks and French fries obtained from them on the content of α-solanine and α-chaconine. The use of tartaric acid in the production of third-generation snacks contributed to an increase in the content of TGA and α-chaconine, regardless of the potato variety. Malic acid turned out to be the most beneficial in their production, regardless of the potato variety and the method of expanding the pellets. Due to the content of glycoalkaloids, a more favourable method of expanding the pellets was frying in deep, hot oil. Third-generation fried snacks contained on average 3 times less glycoalkaloids than those from the microwave, which may be related to the higher dry matter content of potato snacks, which came from the raw material. In the production of French fries, the use of organic acids had little or no effect on the glycoalkaloid content in the finished products and depended on the potato variety. In this respect, it was more advantageous to use potatoes of the MB variety with red flesh and the use of lactic acid and baking fried, frozen semi-finished products rather than deep-frying them in oil. Regardless of the potato variety, the process of baking fried French fries turned out to be more beneficial than frying them due to the three times lower content of glycoalkaloids in the finished products. This was probably related to the greater degradation of glycoalkaloids in a longer process and higher temperature. Studies have shown a greater impact of technological treatments and organic acids on reducing the content of less toxic α-solanine in finished products. With regard to the raw material, 3.31-9.27% TGA remained in third-generation fried snacks, from 11.5 to 23.7% TGA in microwaved snacks, from 8.84 to 34.9% TGA in fried French fries, and from 5.15 to 44.2% TGA in baked fries.

                                                                                                Yours sincerely

                                                                Agnieszka Tajner-Czopek and co-authors

Reviewer 3 Report

Comments and Suggestions for Authors

General comments

The manuscript is well written in general and presents a very interesting and useful study. However, some points need to be improved. The title should be changed to show the whole research and more details are needed in the methodology section.

1.       First of all, I suggest modifying the title to avoid repeated sound in “products produced” and to clarify the “final treatments”. The title is the first approach of the paper to the reader and it is important that it is striking and shows the totality of the study. Thus, the “treatments” need to be at least specified.

2.       Specify at the end of the introduction the type of organic acid and the method of final treatments that will be used in the study.

3.       Please add more information, what was the average size and weight of the samples? I highly recommend including a picture of each variety used.

4.       Add briefly the method to determine the starch and reducing sugar content. The same for anthocyanin content, add column used, injection conditions and main characteristics of the method.

5.       How the obtained dried potatoes in the form of grits were stored?

6.       How were the proportions of the mixture for the extruded pellets selected?

7.       Table 1. I suggest to include the table in vertical alignment, in order to keep the letters for significant differences next to the values.

8.       It is not necessary to repeat within the text the information provided by the table.

9.       Finally, in conclusion section please add more information regarding the importance of the obtained results in the food industry and possibilities of further or complementary analysis.

Author Response

Responses to Reviewer 3 comments

The authors are very grateful to the Reviewer for his time and valuable comments and guidance on the manuscript, which increased its merit and were very helpful. In the responses to the Reviewer and in the manuscript, the amendments has been indicated in blue font.

Reviewer 3

General comments

The manuscript is well written in general and presents a very interesting and useful study. However, some points need to be improved. The title should be changed to show the whole research and more details are needed in the methodology section.

Reviewer's comment 1:  First of all, I suggest modifying the title to avoid repeated sound in “products produced” and to clarify the “final treatments”. The title is the first approach of the paper to the reader and it is important that it is striking and shows the totality of the study. Thus, the “treatments” need to be at least specified.

Response 1:  According the Reviewer suggestion the title of the manuscript has been changed and placed on page 1, in lines from 2 to 4. The title was changed on: “Assessment of the content of glycoalkaloids in potato snacks made from coloured potatoes, resulting from the action of organic acids and thermal processing”.

Reviewer's comment 2: Specify at the end of the introduction the type of organic acid and the method of final treatments that will be used in the study.

Response 2: According the Reviewer suggestion the end of the introduction has been reconstructed and placed on page 2 in lines from 87 to 98.

Due to the need to develop research on the properties of such potatoes an attempt was made to determine the effect of treating tubers with skin with selected food organic acids on the content of glycoalkaloids in dried potato grits, third-generation snacks made from it and in French fries in the process of frying, microwave oven or baking. The research used ascorbic, citric, lactic, malic and tartaric acids, which in pilot studies had a positive effect on the colour of potato products with purple and red flesh.

Therefore, the aim of the research was to determine the influence of the type of organic acid, the method of preparing products for consumption and the type of raw material, i.e. unpeeled potatoes of 2 varieties, differing in the color of the flesh, and thus in the structure and content of anthocyanins, on the content of glycoalkaloids in dried potatoes and third-generation snacks, and fries.

Reviewer's comment 3: Please add more information, what was the average size and weight of the samples? I highly recommend including a picture of each variety used.

Response 3: According the Reviewer suggestion this information has been added and placed in the section 2.1., on page 3 in lines from 105 to 106. The picture of the potatoes has been placed in Supplementary Material as Figure S1.

The size of the potato tubers of MB variety was closed to 60 x 80 mm 105 (width x length) and the tubers of the DF variety to 60 x 90 mm (Figure S1).

Reviewer's comment 4: Add briefly the method to determine the starch and reducing sugar content. The same for anthocyanin content, add column used, injection conditions and main characteristics of the method.

Response 4: According the Reviewer suggestion this information has been added and placed in the section 2.2 on page 3 in lines from 120 to 142.

Concerning the method of starch determining:

The starch content was determined in raw potato tubers indirectly by measuring the specific gravity of tubers and reading the starch value from Maercer’s tables, according to the methodology described by Houghland [25].

Concerning the method of reducing sugars determining:

The reducing sugar were determined by the colorimetric method described by Lindsay [26].

Concerning the method of anthocyanin content determining:

The anthocyanin content was determined using the HPLC method as described below. Freeze-dried potato samples were extracted with 70% aqueous acetone (acidified with 0.1% acetic acid). The homogenized mixture was left for 2 hours at room temperature, and then the acetone-water mixture was separated with chloroform to remove lipophilic compounds. The acetone-water fraction was collected and the remaining acetone was evaporated on a Büchi rotary evaporator. The obtained aqueous extract was brought to a known volume with distilled water and stored at 20°C until analysis. Samples filtered through 0.2 µm filters were used for HPLC analysis of anthocyanins. Anthocyanins were determined using a Dionex HPLC system (USA) equipped with an Ultimate 3000 model diode detector, an LPG-3400A quaternary pump, an EWPS-3000SI automatic sampler and a TCC-3000SD thermostabilized column chamber, controlled by Chromeleon v.6.8 software. An Atlantic T3 reversed-phase column (250 mm x 4.6 i.d., 5 μm) (Waters, Ireland) and an Atlantis T3 guard column (20 x 4.6 i.d., 5 μm) (Waters, Ireland) were used. The following solvents were used as the mobile phase: A/ 4.5% formic acid and B/ acetonitrile. The following elution conditions were used: 0–1 min, 5% B isocratic; 1–6 min, linear gradient from 5–10% B; 6–26 min, linear gradient 10–20% B; 26–33 min, linear gradient from 20–100% B; and then the initial conditions. The flow rate was 1 ml·min-1 and the injection volume was 40 μL. The column was operated at 30°C. Anthocyanins were monitored at a wavelength of 520 nm. 

Reviewer's comment 5: How the obtained dried potatoes in the form of grits were stored?

Response 5: Dried potato grits were stored in the freezer before taking into the experiments. According the Reviewer suggestion the information (These were stored in the freezer before taking into the experiments) has been added on page 4 in line 155.

Reviewer's comment 6: How were the proportions of the mixture for the extruded pellets selected?

Response 6: The proportions of the mixture for the extruded pellets were elaborated previously by the authors Pęksa et al. [27]. According the Reviewer suggestion this information has been added on page 5 in lines from 203 to 204.

Reviewer's comment 7: Table 1. I suggest to include the table in vertical alignment, in order to keep the letters for significant differences next to the values.

Response 7: Table 1 has been placed vertically at the suggestion of the Reviewer - however, in order to increase its readability, it has been divided into 2 separate tables (Table 1. Characteristics of studied potatoes of two varieties and Table 2. Characteristics of experimental potato grits depending on the type of organic acid used). Two tables has been appropriately described in the manuscript (text in blue font).  Table 1 has been placed on page 7 in lines from 298 to 301 and Table 2 – on page 8 in lines from 341 to 346.

Reviewer's comment 8: It is not necessary to repeat within the text the information provided by the table.

Response 8: According the Reviewer suggestion the text has been checked and corrected throughout in the part – Results and discussion.

Changed fragments of the text has been inserted in section 3.1, on page 7, in lines from 317 to 321:

The tested samples of tubers of both potato varieties differed in terms of their total anthocyanin content. Potatoes of the DF variety with purple flesh contained twice as much of these compounds, i.e. on average 50.38 mg·100g-1 FW (Table 1). Lachman et al. [33] reports that potatoes with red flesh contained an average of 231.9 mg·100g-1 DM of anthocyanins, while those with purple flesh contained 146.8 mg·100g-1 DM.

Changed fragments of the text has been inserted in section 3.1, on page 8, in lines from 328 to 330:

….and Nie et al. [35], noted that microwaving and baking processes reduced the amount of glycoalkaloids in unpeeled potatoes by 45% and 51%, respectively compared to the raw material.

Changed fragments of the text has been inserted in section 3.1, on page 9, in lines from 356 to 362:

The lowest TGA content in experimental grits was found in samples obtained from tubers of the MB variety treated with tartaric and citric acid, and from potatoes of the DF variety treated with ascorbic acid. With regard to the raw material, in experimental grits, especially those obtained from potatoes of the DF variety, the content of α-solanine decreased under the influence of blanching and immersion in acid solutions to a greater extent than α-chaconine, with the exception of samples immersed in malic acid (Table 1 and Table 2).

Changed fragments of the text has been inserted in section 3.2, on page 9, in lines from 387 to 389:

…the origin of the grits, the amount of glycoalkaloids in the snacks remained at a low level, α-solanine did not exceed 0.47 mg ·100 g-1, α-chaconine 0.79 mg ·100 g-1, so a total of 1.26 mg ·100 g-1 TGA.

Changed fragments of the text has been inserted in section 3.2, on page 10, in lines from 415 to 421:

…the influence of microwaves, and accordingly, the content of glycoalkaloids in fried snacks from MB variety potatoes reached 0.38 mg ·100 g-1 TGA, and from DF variety potato tubers 0.52 mg ·100 g-1 TGA, while in samples expanded under the influence of microwaves, these values were 1.27 and 1.24 mg ·100 g-1 TGA, respectively. Regardless of the snack expansion method, they contained more α-chaconine than α-solanine. In snacks made from MB potatoes, the proportion of α-chaconine to α-solanine was 1.59, and in products made from DF potatoes it was 1.82 (Figure 2; Table S1).

Changed fragments of the text has been inserted in section 3.2, on page 13, in lines from 486 to 494:

Depending on the type of acid used, glycoalkaloid residues in potato snacks ranged from 11.5 to 15.0% TGA when experimental grits from Mulberry Beauty potato tubers were used and from 22.6 to 23.7% TGA when it was Double Fun potato grits. However, more α-solanine than α-chaconine remained in the snacks.

Research conducted by Rytel et al. [15] showed that the French fries they obtained contained 3%, crisps 16%, and dried potatoes 17% of the initial amount of total glycoalkaloids (TGA) contained in the raw material, and the reduction in the content of α-chaconine and α-solanine was at a similar level.

Changed fragments of the text has been inserted in section 3.3, on page 13, in lines from 486 to 494:

The content of both forms of glycoalkaloids in French fries obtained during the research from potatoes with colored flesh depended primarily on the potato variety and the method of preparing the fries for consumption, and only to a small extent on the type of organic acid used at the stage of blanching the potato strips (Table 4). The interactions of the factors used had a significant impact on the content of glycoalkaloids in ready-made French fries (Figures 3A-C; Table S2).

French fries obtained from tubers of the purple-colored DF variety contained on average 2 times less α-solanine compared to French fries prepared from tubers of the MB variety with red flesh, while the differences between French fries from potatoes of these varieties in terms of α-chaconine and TGA content were minor. A more advantageous method of preparing French fries for consumption in terms of the glycoalkaloid content in the final product was baking a frozen semi-finished product. This concerned the content of α-solanine to a greater extent. Baked French fries contained on average approximately 3 times less α-solanine than fried French fries. Regardless of the type of acid used, the content of α-solanine in the tested French fries decreased compared to the control sample (immersion in water), while the content of α-chaconine and TGA increased, especially after the use of ascorbic acid (Table 4).

Reviewer's comment 9: Finally, in conclusion section please add more information regarding the importance of the obtained results in the food industry and possibilities of further or complementary analysis.

Response 9: According the Reviewer suggestion the information has been improved in the Conclusion – on pages 16-17, in lines from 600 to 628.

Processing potatoes containing anthocyanins requires maintaining the structure and colour of these phenolic compounds and their health-promoting effects. Processing tubers with skin increases the amount of anthocyanins in the products obtained from them, but it is possible to introduce other compounds found in potato skin into the finished products, including glycoalkaloids that are toxic in larger amounts. The research showed a varied impact of the use of 1% solutions of ascorbic, citric, lactic, malic and tartaric acids in the production of dried potato groats and third-generation snacks and French fries obtained from them on the content of α-solanine and α-chaconine. The use of tartaric acid in the production of third-generation snacks contributed to an increase in the content of TGA and α-chaconine, regardless of the potato variety. Malic acid turned out to be the most beneficial in their production, regardless of the potato variety and the method of expanding the pellets. Due to the content of glycoalkaloids, a more favourable method of expanding the pellets was frying in deep, hot oil. Third-generation fried snacks contained on average 3 times less glycoalkaloids than those from the microwave, which may be related to the higher dry matter content of potato snacks, which came from the raw material. In the production of French fries, the use of organic acids had little or no effect on the glycoalkaloid content in the finished products and depended on the potato variety. In this respect, it was more advantageous to use potatoes of the MB variety with red flesh and the use of lactic acid and baking fried, frozen semi-finished products rather than deep-frying them in oil. Regardless of the potato variety, the process of baking fried French fries turned out to be more beneficial than frying them due to the three times lower content of glycoalkaloids in the finished products. This was probably related to the greater degradation of glycoalkaloids in a longer process and higher temperature. Studies have shown a greater impact of technological treatments and organic acids on reducing the content of less toxic α-solanine in finished products. With regard to the raw material, 3.31-9.27% TGA remained in third-generation fried snacks, from 11.5 to 23.7% TGA in microwaved snacks, from 8.84 to 34.9% TGA in fried French fries, and from 5.15 to 44.2% TGA in baked fries.

                                                                                                 Yours sincerely

                                                             Agnieszka Tajner-Czopek and co-authors

Round 2

Reviewer 2 Report

Comments and Suggestions for Authors

All my comments are addressed. Thank you. 

Comments on the Quality of English Language

Minor changes are required. 

Reviewer 3 Report

Comments and Suggestions for Authors

The manuscript was improved, and the authors accepted the suggestions made by the reviewers. Therefore, I accept the manuscript in the current form